# Single Turnover of Transient of Reactants Supports a Complex Interplay of Conformational States in the Mode of Action of *Mycobacterium tuberculosis* Enoyl Reductase

**Leonardo Kras Borges Martinelli** [1], **Mariane Rotta** [1,2], **Cristiano Valim Bizarro** [1] , **Pablo Machado** [1,2] and **Luiz Augusto Basso** [1,2,*]

1 Centro de Pesquisas em Biologia Molecular e Funcional (CPBMF), Instituto Nacional de Ciência e Tecnologia em Tuberculose (INCT-TB), Pontifícia Universidade Católica do Rio Grande do Sul (PUCRS), Porto Alegre 90619-900, Brazil
2 Programa de Pós-Graduação em Medicina e Ciências da Saúde, PUCRS, Porto Alegre 90619-900, Brazil
* Correspondence: luiz.basso@pucrs.br; Tel.: +55-51-33203629

**Abstract:** The enoyl reductase from *Mycobacterium tuberculosis* (*Mt*InhA) was shown to be a major target for isoniazid, the most prescribed first-line anti-tuberculosis agent. The *Mt*InhA (EC 1.3.1.9) protein catalyzes the hydride transfer from the 4$S$ hydrogen of β-NADH to carbon-3 of long-chain 2-*trans*-enoyl thioester substrates (enoyl-ACP or enoyl-CoA) to yield NAD$^+$ and acyl-ACP or acyl-CoA products. The latter are the long carbon chains of the meromycolate branch of mycolic acids, which are high-molecular-weight α-alkyl, β-hydroxy fatty acids of the mycobacterial cell wall. Here, stopped-flow measurements under single-turnover experimental conditions are presented for the study of the transient of reactants. Single-turnover experiments at various enzyme active sites were carried out. These studies suggested isomerization of the *Mt*InhA:NADH binary complex in pre-incubation and positive cooperativity that depends on the number of enzyme active sites occupied by the 2-*trans*-dodecenoyl-CoA (DD-CoA) substrate. Stopped-flow results for burst analysis indicate that product release does not contribute to the rate-limiting step of the *Mt*InhA-catalyzed chemical reaction. The bearings that the results presented herein have on function-based anti-tuberculosis drug design are discussed.

**Keywords:** tuberculosis; InhA; enoyl reductase; mycolic acids; isoniazid; drug target; transient kinetics; stopped flow; single turnover

## 1. Introduction

The Global Tuberculosis Report 2022 published by the World Health Organization (WHO) was based on data reported by 202 countries and territories accounting for more than 99% of the world's population [1]. It has been estimated that approximately one-fourth of the world's population is infected with *Mycobacterium tuberculosis*, the main causative agent of tuberculosis (TB) in humans [1]. Deaths from TB of HIV-negative and HIV-positive patients in 2021 accounted for 1.4 million and 187,000 people, respectively. Moreover, drug resistance continues to be a public health threat [1]. Approximately 85% of people who develop TB can be successfully treated with a 6-month drug regimen for drug-susceptible TB (two months with isoniazid, rifampicin, pyrazinamide and ethambutol, followed by four months with isoniazid and rifampicin). On the other hand, treatment of MDR/RR-TB cases requires a course of second-line drugs, which are more expensive (≥USD 1000 per person), and ought to be supported by counselling and monitoring for adverse events [2]. The development of new oral chemotherapeutic agents is needed to further reduce the course of treatment of TB, and to reduce both the governmental costs of treating TB patients and patient-incurred costs (including travel, nutrition, loss of productivity, physical and mental issues and delayed career growth in young patients).

The completion of the sequencing of the whole *M. tuberculosis* genome [3] offered a springboard from which to launch screening campaigns focused on inhibition of target enzymes via the use of low-molecular-mass chemical compounds. These targets were selected based on gene product essentiality for *M. tuberculosis* growth (and preferentially absence from the human host). The underlying assumption was that in vitro inhibition of enzyme activity of purified protein by chemical compounds with lower overall inhibition constant values would be translated into target specificity and efficacy [4]. However, these efforts have yet to yield new anti-TB agents as target-based in vitro screening of enzyme inhibitors neglects cell wall permeability, metabolic stability and target vulnerability [4]. Accordingly, there has been renewed interest in phenotypic screening in drug discovery efforts to potentially address the incompletely understood complexity of diseases such as tuberculosis and deliver first-in-class drugs [5]. Notwithstanding, it has been pointed out that the most successful campaigns combine target knowledge with functional cellular assays to identify drug candidates with the most advantageous molecular mechanism of action [6]. As pointed out by Nunes et al. [7], among the advantages of the target-based approach to the development of new chemical entities (NCEs) to treat TB, the following ones may be highlighted: (1) availability of recombinant proteins to perform in vitro high-throughput screening campaigns, (2) identification of leads with defined molecular mechanisms against a specific protein target, (3) prompt analysis of compounds with favorable cost/benefit ratios, (4) priority for further development given to compounds with selective toxicity (e.g., enzymes of essential pathways that are absent from the human host), (5) medicinal chemistry efforts can be guided by structural data of the target to, when needed, improve pharmacodynamics and pharmacokinetics, (6) pre-clinical evaluation of lead compounds, (7) lowered attrition rate downstream, as found by drug discovery programs, due to off-target effects and target druggability, (8) the ability to test a combination of compounds early to address drug–drug interactions, persistence and drug resistance, and (9) early filing for intellectual protection if a new chemical scaffold is propounded. Whether a low-molecular-mass chemical compound is derived from target-based drug discovery or phenotypic drug screening, any further efforts to translate a lead into a drug will most certainly benefit from a thorough understanding of the mode of action and target engagement profile.

The mycolyl-arabinogalactan-peptidoglycan complex (mAGP) of mycobacterial envelope is formed by covalent bonds between peptidoglycan, arabinolactan and mycolic acid macromolecules [8]. The latter are high-molecular-weight α-alkyl, β-hydroxy fatty acids that are covalently attached to the hexaarabinosyl of non-reducing termini of the arabinanlactan polymer in the mycobacterial cell wall [9,10]. There are two types of fatty acid synthase systems (FASs) that catalyze β-ketoacyl synthesis, β-ketoacyl reduction, β-hydroxyacyl dehydration and enoyl reduction chemical reactions [8–10]. The FAS-I systems are one-gene-encoded multidomain polypeptides that catalyze these chemical reactions and are present in most eukaryotes, except plants [9]. The FAS-II systems, which are present in bacteria and plants, catalyze these chemical reactions by separate enzymes, each of which is encoded by separate genes [8,10]. The FAS-I and FAS-II systems are present in mycobacteria [8,10]. The mycobacterial FAS-I produces acyl-CoA fatty acids in a bimodal fashion ($C_{14:0}$-$C_{16:0}$ to $C_{24:0}$-$C_{26:0}$) [10]. The shorter acyl-CoA fatty acid precursors ($C_{14:0}$-$C_{16:0}$) produced by the FAS-I system undergo condensation with malonyl-ACP by the mtFabH enzyme, which is further elongated by the FAS-II system to form the meromycolate branch (50–60 carbons) of mycolic acids [10]. The longer-chain acyl-CoA products ($C_{24:0}$-$C_{26:0}$) of FAS-I are substrates for Claisen-type condensation with the meromycolate moiety to form of the α-alkyl branch (20–26 carbons) of mycolic acids [10]. Total sequencing of the *M. tuberculosis* H37Rv genome showed the organization of the open reading frames of the FAS-II system enzymes [3]. In the *inhA* operon, the *inhA* gene that encodes the *trans* Δ2-enoyl reductase enzyme (*Mt*InhA) is located downstream of the *mabA* (also *fabG1* in *M. tuberculosis* or *fabG* in *E. coli*) open reading frame that is transcribed and translated into a β-ketoacyl reductase (MabA) [11]. Cloning, purification and steady-state kinetics

measurements of *Mt*InhA activity showed that it is a β-NADH-dependent enoyl-ACP (acyl carrier protein) reductase enzyme and thus a member of the FAS-II system [12]. *Mt*InhA was also shown to be specific for long-chain ($C_{18} > C_{16}$) enoyl thioester substrates [12]. *Mt*InhA was shown to be a major target for isoniazid (INH) [13–15], which is the most prescribed first-line chemotherapeutic agent for both the treatment of active TB and for prophylaxis of this disease.

The *Mt*InhA (EC 1.3.1.9) enzyme catalyzes the hydride transfer from the 4*S* hydrogen of β-NADH to carbon-3 of long-chain 2-*trans*-enoyl thioester substrates (enoyl-ACP or enoyl-CoA) and induces enolate formation, followed by protonation, yielding NAD$^+$ and acyl (enoyl-ACP or acyl-CoA) products [12,16,17] (Figure 1). Studies on steady-state kinetics suggested that wild-type *Mt*InhA follows a sequential kinetic mechanism in which β-NADH binds first, followed by enoyl-CoA binding to form the catalytically competent ternary complex [12]. On the other hand, primary kinetic deuterium isotope effects on enzyme activity suggested a random-order mechanism of substrate binding to wild-type *Mt*InhA [16]. The latter mechanism has been supported by measurements of quench in intrinsic protein fluorescence at equilibrium and stopped-flow rates for 2-*trans*-dodecenoyl-CoA (DD-CoA) substrate binding to wild-type *Mt*InhA [18]. A thorough description of chemical and kinetic mechanisms, structural data and inhibitors of *Mt*InhA enzyme activity has recently been reported [19]. The graphical abstract of our contribution strives to provide both a summary of past enzyme kinetic data and the conclusions reached by the studies described herein.

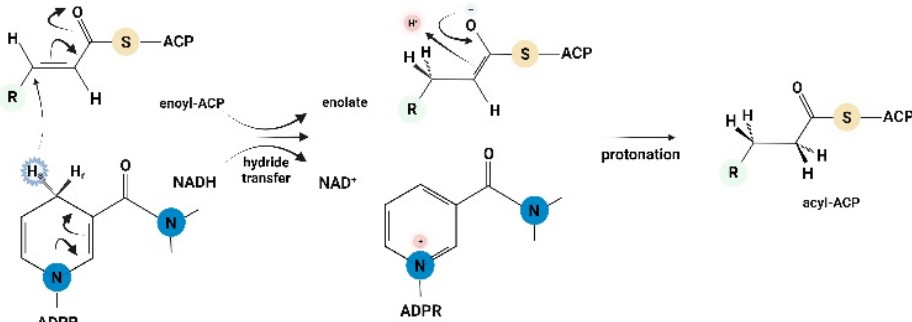

**Figure 1.** *Mt*InhA catalyzes C-4*S* hydride transfer from β-NADH to 2-*trans*-enoyl ACP(CoA), and protonation of enolate intermediate results in formation of NAD$^+$ and enoyl-ACP(CoA) products. Figure drawn using BioRender.com.

The hyperbolic equilibrium binding of NADH to free wild-type *Mt*InhA, which was assessed by measuring the enhancement in nucleotide fluorescence, suggested that the initial nucleotide binding process is followed by a conformational change in the binary complex, yielding an overall dissociation constant value of 0.57 μM [14]. The kinetics of NADH binding to wild-type *Mt*InhA and isoniazid-resistant mutant proteins have been investigated via fluorescence spectroscopy with stopped-flow equipment [20]. The stopped-flow traces showed a biphasic enhancement in nucleotide fluorescence for NADH binding to wild-type and isoniazid-resistant *Mt*InhA mutant proteins [20]. The proposed mechanism included a fast phase for the rapid association of NADH to one conformer of an enzyme and a slow phase that is rate-limited by the slow conversion of two pre-existing conformers of free enzymes in solution [20]. Notwithstanding, the typical hyperbolic decrease in the slow-phase rate constant values with increasing nucleotide concentration could not be observed for wild-type *Mt*InhA [20], as detection of this process also depends on whether or not the equilibrium is displaced to the conformational state of free enzymes that bind NADH. On the other hand, the overall dissociation constant derived from analysis of the total amplitude changes for the fast and slow phases of the stopped-flow biphasic enhancement in nucleotide fluorescence was in agreement with the values obtained from the analysis of rate constants, thereby supporting the proposed mechanism by which free

*Mt*InhA enzymes exist in equilibrium between two conformers and only one of them binds NADH [20].

Binding of the DD-CoA substrate to free *Mt*InhA was monitored by measuring the quench in intrinsic protein fluorescence upon binary complex formation [18]. Equilibrium fluorescence spectroscopy data were sigmoidal, suggesting positive cooperativity upon DD-CoA binding to free enzymes [18]. As studies on steady-state kinetics and equilibrium binding cannot provide insights into isomerization of central complexes or individual rates of substrate binding to free enzymes, stopped-flow measurements of partial reactions or elementary steps were employed to remove these limitations as transient kinetics observe the changes occurring in the enzyme molecule itself and thus help clarify the elementary steps of enzyme reaction. Stopped-flow traces of quench in intrinsic protein fluorescence for *Mt*InhA:DD-CoA binary complex formation were biphasic [18]. The observed apparent rate for the fast phase showed a linear dependence on increasing DD-CoA concentration, suggesting a single-step reversible bimolecular association process. On the other hand, the observed apparent rate for the slow phase displayed a hyperbolic increase, suggesting a bimolecular association process followed by a slow unimolecular isomerization of the *Mt*InhA:DD-CoA binary complex [18].

On one hand, the studies on NADH binding suggested two forms of free *Mt*InhA in solution prior to nucleotide binding, followed by a conformational change in the binary complex with no cooperativity. On the other hand, enoyl-CoA binding results suggest a bimolecular association process followed by a slow isomerization of the binary complex and that there is positive homotropic cooperativity. There are two generally invoked allosteric mechanisms for sigmoidal binding data: (1) the symmetry or concerted model, which predicts that two forms of free enzymes exist at equilibrium in solution [21], and (2) the sequential or induced-fit model, which suggests one form of free enzymes in solution and a slow isomerization of the binary complex upon ligand binding to the enzyme [22]. The isomerization steps are necessarily slower than the ligand binding steps in both the symmetry and sequential models [21,22]. The symmetry model for cooperative transitions can be divided into two systems: (1) K systems, in which the transition from low to high or from high to low affinity is triggered by substrate binding to an allosteric site, and (2) V systems, in which the substrate has the same affinity for the two states and acceleration or deceleration can be observed only if the two states of the protein differ in their catalytic activity, and the substrate can act as an effector if binding to an allosteric site of active (positive V system) or inactive (negative V system) state occurs [21]. Whether or not free *Mt*InhA exists in two interconvertible forms and only one of them binds NADH (symmetry model), in which no cooperativity can be observed at equilibrium, or the enoyl-CoA substrate binds to one form of free enzymes in solution followed by binary complex conformational change with positive cooperativity (induced-fit model) observed at equilibrium, remains to be solved. It has long been recognized that records of time courses following either the formation of products or detection of transient intermediates during the first turnover of enzymes are uniquely suited for showing either the independence of sites or interactions between sites of oligomeric polypeptides (quaternary structure) during catalysis [23]. Thus, the equivalence or non-equivalence of the enzyme active sites may be investigated by measuring rates of product formation under conditions in which enzyme active-site concentration is in excess over substrate concentrations (single-turnover experiments) with varying degrees of occupancy of the enzyme active sites of oligomeric proteins. For instance, single-exponential records, regardless the degree of site occupancy, suggest a lack of interaction between the enzyme active sites. It appears appropriate to distinguish between the transient of reactants and the transient of intermediates [23]. Following the time course of total concentration of product (enzyme-bound and free) informs on the transient of reactants, thereby yielding purely kinetic information [23]. Detecting the changes in concentration of the different forms of enzymes (free enzyme, enzyme substrate and enzyme product complexes) informs on the transient of intermediates [23]. Accordingly, the single-turnover data presented herein are those of studies on the transient of reactants. The relevance of our contribution

to the target-based rational design of functional inhibitors of *Mt*InhA enzyme activity as potential anti-TB agents is discussed in Section 4 (Conclusions).

## 2. Materials and Methods

### 2.1. Recombinant MtInhA Expression and Purification and Enoyl-CoA Substrate Synthesis and Purification

The recombinant *Mt*InhA was expressed and purified as previously described [14,24]. The 2-*trans*-dodecenoyl-CoA (DD-CoA) substrate was synthetized from 2-*trans*-dodecenoic acid and coenzyme A via anhydride formation following acylation [14], and purified via reverse-phase HPLC using a $19 \times 300$ mm $C_{18}$ μBondapak column (Waters Associates, Milford, MA, USA) as previously described [16].

### 2.2. Single-Turnover Experiments

The decrease in NADH concentration upon double-bond reduction of the DD-CoA substrate catalyzed by *Mt*InhA was analyzed in single-turnover conditions in the Applied-Photophysics SX-18MV-R stopped-flow instrument (Applied Photophysics Ltd., Leatherhead, UK), measuring absorbance at 340 nm at 25 °C. The time courses for all experiments were 500 ms, using 400 data points for each time base and an optical path of 10 mm. All concentration values for the enzyme and substrates reported herein are for mixing chamber concentrations. In one of the single-turnover experiments, 15 μM of *Mt*InhA tetramers was mixed with either 5, 10 or 14 μM of NADH and 225 μM of DD-CoA with no pre-incubation of either the substrate or the enzyme. Another experiment involved pre-incubation for 10 min of tetrameric *Mt*InhA (15 μM) with NADH (5, 10 or 14 μM of NADH) and the subsequent addition of DD-CoA (225 μM). An experiment was also carried out in which 15 μM of *Mt*InhA tetramers was pre-incubated for 10 min with DD-CoA (14 μM, 30 and 39 μM) before being mixing with NADH (400 μM). The chemical control experiments were carried out with the same concentrations of substrates and omission of *Mt*InhA. Note that the excess concentration values for DD-CoA and NADH were chosen to be larger their Michaelis–Menten constants, which are the following: NADH ($K_m \cong 66$–$113$ μM) and DD-CoA ($K_m \cong 48$–$57$ μM) [12,17,24]. The stopped-flow traces shown are for the averages of five individual reactions of *Mt*InhA for all single-turnover experiments.

### 2.3. Burst

To determine whether product release is part of the rate-limiting step, pre-steady-state kinetic measurements of the reaction catalyzed by *Mt*InhA were performed using an Applied Photophysics SX 18MV-R stopped-flow spectrofluorometer on absorbance mode. The decrease in absorbance was monitored at 340 nm (1 mm slit width = 4.65 nm spectral band) and 25 °C, using a time course of 500 ms, 400 data points for each time base and an optical path of 10 mm. The experimental conditions were 10 μM *Mt*InhA, 300 μM NADH and 225 μM DD-CoA in 100 mM Pipes pH 7.0 (mixing chamber concentrations). The experimental conditions for the control experiment were 300 μM NADH and 225 μM DD-CoA in 100 mM Pipes pH 7.0 (mixing chamber concentrations). The dead time of the stopped-flow equipment was 1.37 ms. The stopped-flow trace shown represents the average of ten individual reactions of *Mt*InhA.

## 3. Results and Discussion

The subheadings presented below strive to provide a concise description of the experimental results, data analysis and interpretation for the transient of reactants under single-turnover experimental conditions and for burst in product formation.

### 3.1. Single Turnover

The decrease in NADH concentration upon double-bond reduction of 2-*trans*-dodecenoyl-CoA (DD-CoA) by *Mt*InhA under single-turnover conditions was followed in the stopped-flow instrument. The large excess of the second substrate (either DD-CoA or NADH) was

employed to ensure that binding to the binary complex would not play any role in limiting enzyme catalysis, which should hopefully simplify the reaction model.

A quadratic equation (Equation (1)) was used to estimate the occupancy of *Mt*InhA substrate binding sites. Equation (1) represents the solution of a quadratic equation for a simple binding process [25].

$$[ES] = \frac{([E_0] + [S_0] + K_D) \pm \sqrt{([E_0] + [S_0] + K_D)^2 - 4([E_0][S_0])}}{2} \tag{1}$$

Equation (1) was employed to provide, for instance, the *Mt*InhA:NADH binary complex ((ES)) concentration using the overall dissociation constant value of 2 μM (as described in the Results and Discussion section), and mixing 15 μM of *Mt*InhA tetramers with either 5, 10 or 14 μM of NADH (as described in the Materials and Method section). Hence, as the quadratic equation provides the concentration of the binary complex, it is just a matter of dividing the binary complex concentration by total enzyme concentration and multiplying this value by 100 {([MtInhA:NADH]/[MtInhA]) × 100}. The NADH concentrations were chosen to obtain an increasing number of monomers of tetrameric *Mt*InhA enzyme active sites occupied by this substrate ($K_D$ = 2 μM) [12]. Namely, 5 μM equals one monomer, 10 μM two monomers and 14 μM three monomers of tetrameric *Mt*InhA occupied by NADH (Table 1).

**Table 1.** Results for transient of reactants. The mixing chamber *Mt*InhA enzyme concentration was 15 μM.

| Single Turnover (No Incubation) | | |
|---|---|---|
| (NADH) (Active Sites) | $k_{obs}$ | |
| (5 μM) 28% | $9.3 \pm 0.1$ s$^{-1}$ | |
| (10 μM) 52% | $10.6 \pm 0.03$ s$^{-1}$ | |
| (14 μM) 66% | $11.3 \pm 0.01$ s$^{-1}$ | |
| Single turnover (NADH incubation) | | |
| (NADH) (active sites) | $k_{obs1}$ | $k_{obs2}$ |
| (5 μM) 28% | $87.2 \pm 3.7$ s$^{-1}$ | $9.1 \pm 0.2$ s$^{-1}$ |
| (10 μM) 52% | $72.5 \pm 1.1$ s$^{-1}$ | $7.0 \pm 0.2^{-1}$ |
| (14 μM) 66% | $52.6 \pm 0.5$ s$^{-1}$ | $14.7 \pm 0.1$ s$^{-1}$ |
| Single turnover (DD-CoA incubation) | | |
| (DD-CoA) (active sites) | $k_{obs}$ | |
| (14 μM) 14% | $6.8 \pm 0.03$ s$^{-1}$ | |
| (30 μM) 55% | $7.6 \pm 0.02$ s$^{-1}$ | |
| (39 μM) 65% | $10.7 \pm 0.02$ s$^{-1}$ | |

The pre-steady-state time course of decreasing absorbance at 340 nm upon NADH conversion to NAD$^+$ was fitted to a single-exponential decay equation (Equation (2)), in which *A* is the absorbance at time *t*, $A_0$ is the absorbance at time zero, and $k_{obs}$ is the apparent first-order rate constant for product formation (Figure 2). The $k_{obs}$ values for each NADH concentration are given in Table 1. Here, it is assumed that conversion of substrates to products goes to completion.

$$A = A_0 e^{-kt} \tag{2}$$

As pointed out above, single-turnover experiments can be used to test whether the sites of oligomeric enzymes are kinetically identical. The results for the single-turnover experiment without incubation show a linear relationship between $k_{obs}$ values and an increasing number of *Mt*InhA sites occupied by NADH (Figure 3). These results suggest that the active sites of tetrameric *Mt*InhA are kinetically equivalent, which is also borne out by the single-exponential records for any degree of site occupancy. However, it is also

possible that NADH binding occurs after DD-CoA substrate binding to form the ternary complex and the ensuing turnover as there was no pre-incubation of enzymes with a reduced dinucleotide substrate.

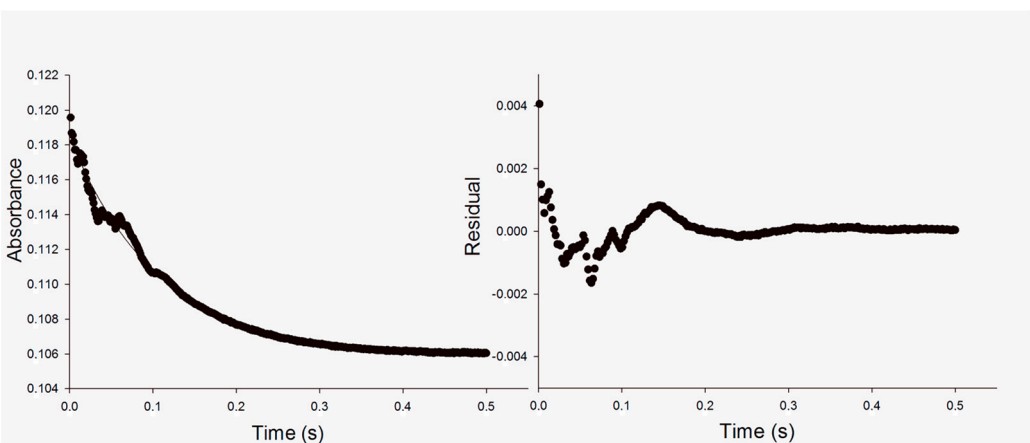

**Figure 2.** Left panel: representative stopped-flow trace of *Mt*InhA (15 μM) reaction showing decrease in absorbance at 340 nm and 25 °C upon conversion of NADH (14 μM) and DD-CoA (225 μM) into NAD$^+$ and dodecanoyl-CoA over the course of the experiment (left panel). The pre-steady-state time course was fitted to a single-exponential decay equation (Equation (2)). Note: all concentration values for enzymes and substrates are those used for mixing chamber concentrations. Right panel: the calculated residual values plotted against time (s) using Equation (2).

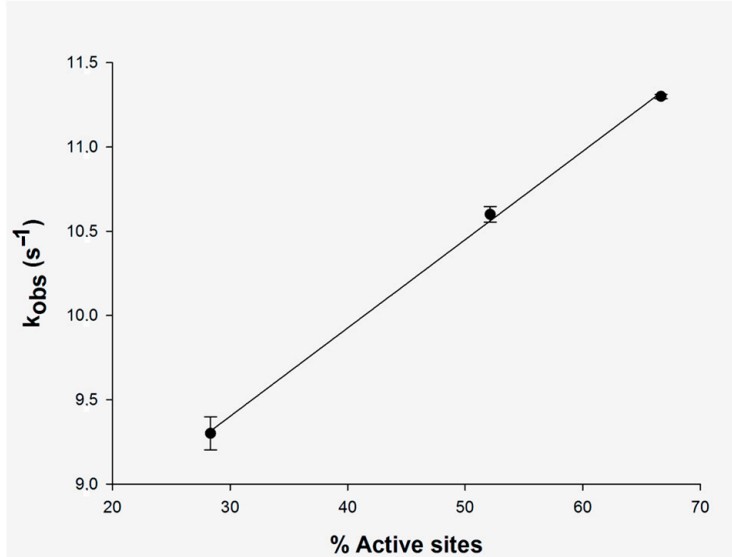

**Figure 3.** Linear dependence of $k_{obs}$ values on the degree of occupancy of active sites of *Mt*InhA (15 μM) without pre-incubation with NADH concentrations of 5 μM (28% occupancy), 10 μM (52% occupancy) and 14 μM (66% occupancy). The $k_{obs}$ values were derived from the best fit of the data of Figure 2 to a single-exponential function, yielding the standard error represented by the bars in the plot.

To try remove this ambiguity, another experiment was carried out in which the enzyme *Mt*InhA (15 μM of tetramers) was pre-incubated with NADH (5, 10 or 14 μM) for 10 min before starting the reaction with a large excess of DD-CoA (225 μM) in the stopped-flow instrument (mixing chamber concentrations). The pre-steady-state time course of the reaction with the pre-incubation of *Mt*InhA and NADH was fitted to a double-exponential

function equation (Equation (3)), yielding values for the apparent rate constant for the fast ($k_{obs1}$) and slow ($k_{obs2}$) phases of the reaction.

$$A = A_1 e^{-k_1 t} + A_2 e^{-k_2 t} \qquad (3)$$

The results for the number of active sites of tetrameric enzymes occupied by NADH prior to catalysis are given in Table 1. The $k_{obs1}$ decreased as a function of increasing NADH concentration (Table 1, Figure 4). These data indicate an isomerization step between two forms of the *Mt*InhA:NADH binary complex in solution and binding of DD-CoA to one of the conformers [20]. The fairly constant values for $k_{obs2}$ (Figure 4) are in agreement with the $k_{cat}$ values (9 s$^{-1}$) for the *Mt*InhA enzyme [24]. Notwithstanding, the $k_{obs1}$ values are larger than $k_{cat}$ and hence appear not to play a role in limiting the enzyme-catalyzed chemical reaction. Interestingly, the double-exponential traces suggest that the active sites are not equivalent, which is in agreement with the dependence of $k_{obs1}$ on the degree of site occupancy, suggesting an equilibrium in the solution of two conformers of the *Mt*InhA:NADH binary complex. The studies on the transient of reactants described herein are in agreement with the hyperbolic NADH binding at equilibrium [14] and the pre-steady-state transient found in intermediate studies [20].

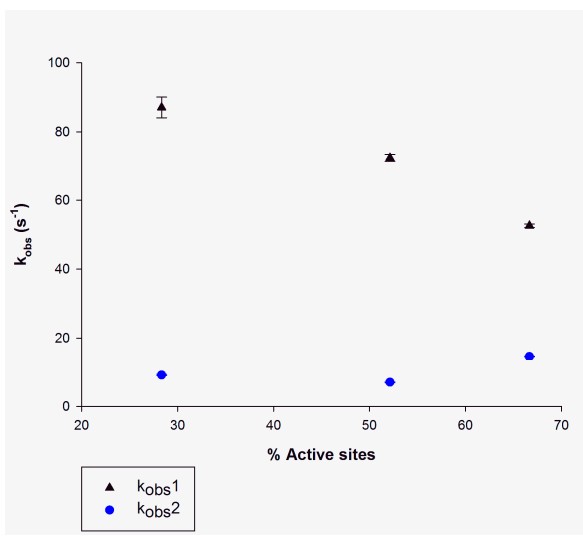

**Figure 4.** The dependence of k$_{obs}$ values derived from data fitting to a double-exponential function (Equation (3)) on the degree of occupancy of active sites of *Mt*InhA (15 µM) pre-incubated with NADH concentrations of 5 µM (28% occupancy), 10 µM (52% occupancy) and 14 µM (66% occupancy). The (▲) represents the $k_{obs1}$ for the fast phase of the biphasic reaction (in black), and (●) represents the $k_{obs2}$ for the slow phase of the biphasic reaction (in blue). The bars in the plot represent the standard errors of data fitting to Equation (3).

For the single-turnover experiments with *Mt*InhA pre-incubated with a DD-CoA substrate in single-turnover kinetics, attempts were made to fit the data to the double-exponential decay equation, as was performed for NADH. However, single and double equations yielded statistically indistinguishable results. The values for $k_{obs1}$ and $k_{obs2}$ for the double equation were fairly similar (results not shown). Accordingly, single-exponential decay (Equation (2)) was employed, and the $k_{obs}$ values are given in Table 1. The results demonstrate a non-linear increase in $k_{obs}$ as a function of increasing DD-CoA concentration (Figure 5). Although the single-exponential traces indicate equivalent binding sites, the non-linear dependence of $k_{obs}$ suggests positive cooperativity that depends on the number of enzyme sites occupied by DD-CoA that increase the binding and/or turnover rate upon NADH association with the *Mt*InhA:DD-CoA binary complex. A slow isomerization of the *Mt*InhA:DD-CoA binary complex prior to NADH binding to form the ternary complex and induce catalysis can be ruled out as $k_{obs}$ values would decrease as a function

of increasing active site occupancy. The results presented herein are in agreement with positive cooperativity upon DD-CoA binding to free *Mt*InhA detected by fluorescence spectroscopy at equilibrium (K' = 8.2 μM; *n* = 2) [18]. Moreover, the studies on the transient of reactants described herein are also in agreement with those in which analysis of total stopped-flow signal amplitude showed that the changes in biphasic quench in protein fluorescence for binary complex formation were sigmoidal (K' = 14.1 μM; *n* = 2.5), and isomerization of the *Mt*InhA:DD-CoA binary complex was proposed [18]. Unfortunately, the experimental conditions precluded the possibility of achieved higher subunit occupancy and thus providing a more thorough description of the dependence of $k_{obs}$ on DD-CoA concentration. At any rate, data for the initial part of a sigmoidal increase in $k_{obs}$ as a function of DD-CoA appear to have been captured (Figure 5).

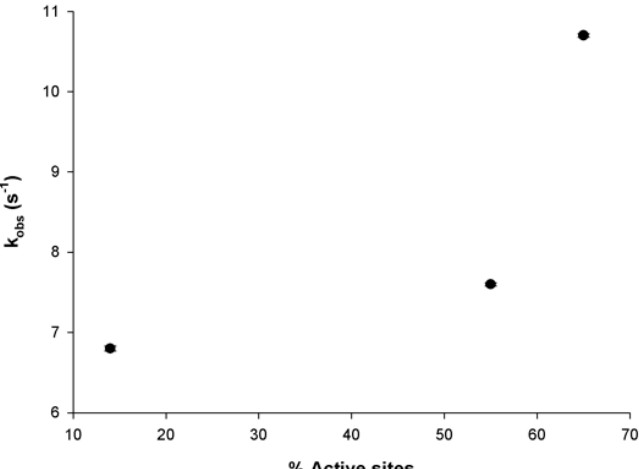

**Figure 5.** The dependence of $k_{obs}$ values on the degree of occupancy of active sites of *Mt*InhA with substrate DD-CoA. The bars in the plot represent the errors.

*3.2. Burst*

Transient kinetics of the forward reaction catalyzed by *Mt*InhA at large concentrations of enzymes and substrates were performed to determine whether product release is part of the rate-limiting step of catalysis. The pre-steady-state time course of the reaction (Figure 6) was fitted to an equation describing single-exponential decay (Equation (2)). This analysis yielded a value of $0.680 \pm 0.003$ s$^{-1}$ for the apparent first-order rate constant. The value of 0.68 s$^{-1}$ for the change in absorbance at 340 nm for the exponential decay in the pre-steady-state experiment for the *Mt*InhA corresponds to 3.8 s$^{-1}$ (using $\Delta \varepsilon$ 6220 M$^{-1}$ cm$^{-1}$, optical path 1 cm and *Mt*InhA subunit concentration of 10 μM). This result is in reasonably good agreement with the $k_{cat}$ determined by the initial velocity experiment in steady-state kinetics, which ranged from 2.8 s$^{-1}$ in 30 mM PIPES pH 6.8 at 25 °C [12] and 4.6 s$^{-1}$ in 30 mM PIPES 150 mM NaCl pH 6.8 at 25 °C [16] to 9 s$^{-1}$ in 100 mM Pipes 100 mM pH 7.0 at 25 °C [24]. The observation of burst during a time course in the pre-steady state phase of the reaction is evidence of a build-up of product in the active site prior to being released into solution. If a burst is observed during the transient phase, and the concentration of NAD$^+$ produced is approximately equal to the enzyme subunit concentration at time zero, it would indicate that the product release is a slower step compared to the chemical step of the reaction [26]. The stopped-flow results indicate that product release does not contribute to the rate-limiting step of the *Mt*InhA-catalyzed chemical reaction as no burst in product formation could be detected (Figure 6).

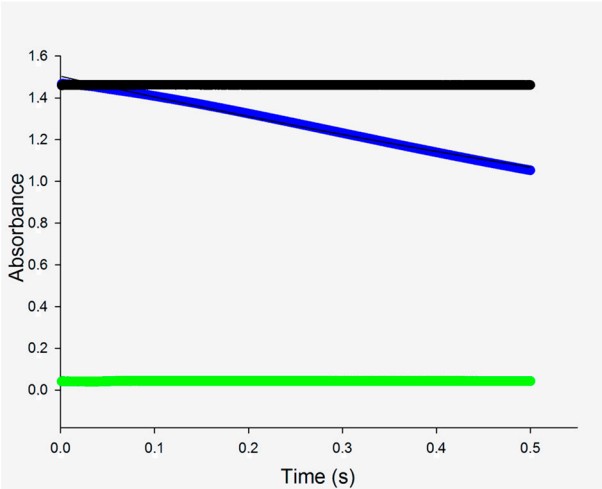

**Figure 6.** Stopped-flow trace for substrate consumption by measuring the decrease in absorbance at 340 nm at 25 °C for *Mt*InhA (10 μM). Data were fitted to Equation (2) (single-exponential decay). The upper time course (in black) is for negative chemical control in which 100 mM Pipes pH 7.0 buffer in one syringe was mixed with NADH (600 μM) and DD-CoA (450 μM) in another syringe (absence of enzyme). The lower time course (in green) is for mixing enzyme (20 μM) in one syringe mixed with 100 mM Pipes pH 7.0 buffer in another syringe (absence of substrates). The time course shown in blue is for the chemical reaction upon mixing *Mt*InhA (20 μM) in one syringe with NADH (600 μM) and DD-CoA (450 μM) in another syringe. Note that the mixing chamber concentrations are 10 μM *Mt*InhA, 300 μM NADH and 225 μM DD-CoA.

## 4. Conclusions

The process by which biological macromolecules (mostly proteins) transmit the effect of binding at one site to another, often distal and functional site, is referred to as allostery, which is important for the regulation of biological activity [27]. As the experimental technologies have improved in sophistication, the concepts and models of allostery have evolved. The concept of allostery evolved from the pre-allostery Bohr effect to allostery from conformational change in a two-state model (1965, 1966), dynamic allostery (1984), ensembles of multiple states (1999), energetic connectivity between residues of proteins and allosteric networks, to a unified view including thermodynamics, population shift and three-dimensional structure [28]. The largely qualitative, static images of end point protein structures (e.g., X-ray crystallography to provide structural information of the protein before and after perturbation) have been replaced with more quantitative, dynamic views of allostery [27]. Computational and experimental methods to study protein dynamics provide access to protein shape-shifting processes that may reveal mechanisms of allosteric communication and features such as cryptic pockets [29]. These studies may offer new therapeutic opportunities, including a better understanding of biological systems and diseases, and informing allosteric drug design efforts [30]. These opportunities include, but are not limited to, enhancement (rather than inhibition) of protein function, displacement of equilibrium to lower substrate affinity conformers, avoidance of off-target effects of inhibitors of function of proteins belonging to large families that show a high degree of conservation (e.g., kinases, NAD(P)-dependent enzymes) and targeting functional sites that are apparently undruggable (e.g., protein–protein interactions, flat and extensively solvent-exposed sites) [29]. Allosteric sites for drug discovery serve as an alternative to enzyme active sites since their lower conservation may be translated into higher selectivity, lower metabolic interference and lower undesirable off-targets [29]. Moreover, allosteric sites can offer new opportunities for optimization of pharmacokinetic properties whenever it is needed [29]. These objectives are thought to be difficult to achieve because the rational-based approach is currently limited to employing intermolecular interactions between

ligands and key amino acids to sterically occlude key functional sites to inhibit undesirable biological activities [29].

An example for tuberculosis includes a synthetic azetidine derivative that is an allosteric inhibitor of tryptophan synthase (an allosterically regulated enzyme) and shown to kill *M. tuberculosis* [31]. These authors suggested that, as the inhibitor binds to α-β-subunit interface of tryptophan synthase and affects multiple steps in the overall enzyme-catalyzed chemical reaction, the resulting inhibition would not be overcome by changes in the metabolic environment [31]. This study also highlights the effectiveness of allosteric inhibition for dynamic protein targets that are essential in vivo despite being dispensable under in vitro conditions [31]. The single-turnover results described herein suggest that *Mt*InhA displays a typical symmetry model [21] with free enzymes in equilibrium between two conformers, followed by NADH binding to one conformer and conformational change in this binary complex. On the other hand, a sequential model [22] with only one conformer of free *Mt*InhA in solution to which DD-CoA binds with homotropic positive cooperativity was invoked. Incidentally, molecular dynamics simulations have shown a normal distribution of free *Mt*InhA enzymes in closed and open conformations [24]. Moreover, classical molecular dynamics simulations of tetrameric *Mt*InhA showed protein flexibility that was borne out by the conformational space sampled by apo, binary and ternary complex forms of enzymes [32]. Molecular dynamics has been employed to study the influence of restrictions of structural flexibility on the A loop (F97-H121) and B loop (D150-A167) of the substrate-binding pocket to propose a monomeric structure that mimics the biologically active tetrameric *Mt*InhA enzyme [33]. These efforts were made to propose a restrained force constant monomeric model system that describes the flexibility of the quaternary structure of *Mt*InhA to evaluate ligands identified via virtual screening without losing dynamics information at a lower computational cost [33]. Future efforts to develop *Mt*InhA enzyme inhibitors may include modeling of ligands to either the protein:DD-CoA binary complex or free enzymes with a larger dissociation constant for the formation of the protein:NADH binary complex.

**Author Contributions:** Conceptualization, M.R., L.K.B.M., P.M., C.V.B. and L.A.B.; Methodology, M.R. and L.K.B.M.; Investigation, M.R. and L.K.B.M.; Writing—Original Draft Preparation, M.R. and L.K.B.M.; Writing—Review and Editing, M.R., L.K.B.M., C.V.B., P.M. and L.A.B.; Supervision, C.V.B., P.M. and L.A.B.; Funding Acquisition, C.V.B., P.M. and L.A.B. All authors have approved the submitted version and agreed to be personally accountable for their contribution and questions related to the accuracy and integrity of this work. All authors have read and agreed to the published version of the manuscript.

**Funding:** P.M.: C.V.B. and L.A.B. would like to acknowledge financial support from the National Institute of Science and Technology on Tuberculosis (INCT-TB) given by Decit/SCTIE/MS-MCT-CNPq-FNDTC-CAPES (grant number 4217-03/2017-2), FAPERGS (grant numbers 17/1265-8 INCT-TB and 19/1724-3 PQG) and BNDES (grant number: 14.2.0914.1). L.K.B.M. acknowledges a post-doctoral fellowship awarded by INCT-TB (CNPq). M.R. acknowledges a doctoral scholarship awarded by CNPq. L.A.B. (CNPq, grant 303499/2021-4), P.M. (305203/2018-5) and C.V.B. (311949/2019-3) are research career awardees of the National Council for Scientific and Technological Development of Brazil (CNPq). This study was financed in part by the Coordination for the Improvement of Higher Education Personnel (CAPES) of Brazil—Finance Code 001.

**Institutional Review Board Statement:** Not applicable.

**Informed Consent Statement:** Not applicable.

**Conflicts of Interest:** The authors declare no conflict of interest.

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
