# Peer review of "Single Turnover of Transient of Reactants Supports a Complex Interplay of Conformational States in the Mode of Action of Mycobacterium tuberculosis Enoyl Reductase"

_futurepharmacol, doi:10.3390/futurepharmacol3020023_

Round 1

Reviewer 1 Report

The work studied the binding kinetics among NADH, DD-CoA and the enoyl reductase from Mycobacterium tuberculosis (MtInhA), an important target for anti-tuberculosis agent. With stopped-flow experiments under single-turnover experimental conditions, the binding kinetics were measured at various enzyme active site occupancy. The manuscript is well organized and the data can largely support the conclusions. The manuscript is recommended to be publishable except the following issues.

(1) page 4, line 202, ‘The’ should be ‘the’ and ‘tha’ should be ‘than’.

(2) “The fairly constant values for kobs2” in Figure 3 is not obvious, as the differences among kobs2 are larger than the ones of kobs in case of single turnover without incubation.

Author Response

REVIEWER #1

We, the authors, were pleased that reviewer#1 was of the following opinion: The manuscript is well organized and the data can largely support the conclusions. The manuscript is recommended to be publishable except the following issues.” This reviewer, however raised minor issues that should be addressed. The issues raised by this reviewer and our responses to them are as follows:

Issue #1: “(1) page 4, line 202, ‘The’ should be ‘the’ and ‘tha’ should be ‘than’.”

Response: Thanks for bringing it to our attention. This sentence is now as follows (page 4, lines 178-180): “The isomerization steps are necessarily slower than the ligand binding steps in both the symmetry and sequential models [21, 22].”

Issue #2: “The fairly constant values for kobs2” in Figure 3 is not obvious, as the differences among kobs2 are larger than the ones of kobs in case of single turnover without incubation.”

Response: Please, note that the values given in Table 1 for kobs without incubation with NADH (9.3, 10.6 and 11.3 s-1) are somewhat similar to kobs2 for pre-incubation of MtInhA enzyme with varying concentrations of NADH (9.1, 7.0 and 14.7 s-1). These values are likely reflecting the catalytic rate for MtInhA enzyme (kcat = 9 s-1). It should, however, be pointed out these data have to be considered with caution as larger than 66% of site occupancy could not be achieved due to protein solubility and foaming in the mixing chamber, which ought to result in experimental artefacts and hence not reliable data. Incidentally, Figure 3 of previous version is now Figure 4 of the current version of our manuscript.

Reviewer 2 Report

In this article, Martinelli et al characterize the single turnover kinetics of M. tuberculosis enoyl reductase (MtInhA) under various substrate and product conditions. Through these conditions, they provide further support for the various conformational states of MtInhA and how its two substrates preferentially bind to individual conformational states. The results of these experiments are complex and multi-faceted and seem appropriate. However, the figures presenting the results need revision to help clarify their outcomes. The article also reads more like a review article with only minor new results presented, which makes it difficult to follow the essential background necessary to interpret the results and to understand where these results fit into previous studies. The authors need to either add more figures and formatting structure to help organize the extraneous background material and conclusions or they need to pare down the article to focus on only the relevant background. With the revisions outlined below, the article could be ready for publication in Future Pharmacology.

1.     The introduction reads more like a review article over M. tuberculosis biology, drug treatment, cell wall synthesis, and eventually MtInhA mechanism and kinetics than a focused background to this article. This reviewer sees two potential outcomes to this current format:

a.     The authors keep this amount of background and add some figures and formatting to help the reader follow along with this review material. Specifically, multiple figures should be added to the background to support the written description. For example, a complete rundown of the mycobacterial envelope is provided with introductions to all of the proteins, lipids, and cofactors involved in this production. The authors need to add a cartoon image of this overall pathway with subsections highlighting the key subpathways that MtInhA is directly involved in. Additionally, the authors should add a chemical scheme showing the chemical reaction performed by MtInhA with chemical structures of the multiple substrates/products. Adding information about the MtInhA mechanism and what is already known about its kinetics in graphical form would also help interpret their data in relation to these past results. A clearer statement of their direct hypothesis would also be helpful. For formatting, the authors should add subheadings to break up these multiple introduction topics.

b.     The authors significantly trim back the extraneous background on M. tuberculosis biology, drug treatment, and cell wall synthesis. A one paragraph summary of MtInhA and its role in M. tuberculosis biology would probably suffice. In this shortened version, the authors should still provide the chemical scheme for the MtInhA reaction and a graphical representation of the MtInhA mechanism and summary of past kinetic findings about MtInhA. This will help with interpretation of their results in either format.

2.     Similar to the introduction, the conclusion section deviates far from the main story of the results from this article discussing allostery, historical interpretation of allostery, cryptic allosteric sites, computer programs to find cryptic sites, and an unrelated allosteric inhibitor of tryptophan synthase from M. tuberculosis. Unlike the introduction where the extraneous information is mostly harmless to the science presented, this winding conclusion section makes it very difficult to figure out what the major conclusions are from their results. For example, the long discussion of cryptic sites and allosteric inhibitors is irrelevant to MtInhA as the authors state in lines 448-449 that no cryptic sites have been identified in MtInhA. This conclusion section needs to be completely revised and refocused on putting their results in context with the previous kinetic findings. A figure to summarize their results, especially in relation to their new introductory figure summarizing past kinetic findings about MtInhA, would really help understand their kinetic results.

3.     The presentation of figures in the article and accompanying figure legends need to be revised to make it easier to interpret their experimental results.

a.     Figure 1: Both figures are too small to read. Text and plotted points need to be made larger. The figure legend needs significant expansion. What data are presented in this figure? What substrate is tested? What are the experimental values for enzyme and substrate concentrations used in this experiment? What is the line shown on the major figure? A more complete descriptive figure legend needs to be added.

b.     Figure 2: Text should be made larger. For the figure legend, what type of error is shown? What are the experimental conditions?

c.     Figure 3: Text and symbols should be made larger. Different colors for the two different data sets would more clearly differentiate them. What type of error is plotted? What are the experimental conditions?

d.     Figure 5: Add color to more clearly differentiate the various conditions. Add experimental details about the conditions used in this experiment.

e.     Building on the introductory figure summarizing past kinetic analyses of MtInhA, the authors should add a cartoon figure into their results section summarizing their various hypotheses about MtInhA structure and conformational change and how to interpret these hypotheses in relation to their experimental data.

4.     Add some more background descriptions to the results and discussion section

a.     How did the authors decide on a quadratic equation (eq 1) for the binding sites on MtInhA? The authors do not justify this choice.

b.     How did the authors decide on their concentrations of NADH to bind to varying occupancies of monomers? (Lines 246-247). The origination of these values is unclear.

c.     Similarly in Table 1, the authors present calculations for the percent of active sites occupied. Where do these values come from? Are these calculated values or are they experimentally determined?

d.     Lines 284-285: How do the authors come to the conclusion about the isomerization step and DD-CoA binding based on the data presented in Figure 3? The authors need to explain this data interpretation more thoroughly.

5.     Minor details

a.     Multiple spelling and grammatical errors in introduction. Please check carefully.

b.     Reference 1 is blank.

Author Response

REVIEWER #2

We, the authors, were glad that reviewer #2 found that “The results of these experiments are complex and multi-faceted and seem appropriate.” This reviewer was of opinion that “With the revisions outlined below, the article could be ready for publication in Future Pharmacology.” Our responses to the issues raised and suggestions made are as follows:

Comments and Suggestions for Authors

In this article, Martinelli et al characterize the single turnover kinetics of M. tuberculosis enoyl reductase (MtInhA) under various substrate and product conditions. Through these conditions, they provide further support for the various conformational states of MtInhA and how its two substrates preferentially bind to individual conformational states. The results of these experiments are complex and multi-faceted and seem appropriate. However, the figures presenting the results need revision to help clarify their outcomes. The article also reads more like a review article with only minor new results presented, which makes it difficult to follow the essential background necessary to interpret the results and to understand where these results fit into previous studies. The authors need to either add more figures and formatting structure to help organize the extraneous background material and conclusions or they need to pare down the article to focus on only the relevant background. With the revisions outlined below, the article could be ready for publication in Future Pharmacology.

Issue: 1. The introduction reads more like a review article over M. tuberculosis biology, drug treatment, cell wall synthesis, and eventually MtInhA mechanism and kinetics than a focused background to this article. This reviewer sees two potential outcomes to this current format:

a.The authors keep this amount of background and add some figures and formatting to help the reader follow along with this review material. Specifically, multiple figures should be added to the background to support the written description. For example, a complete rundown of the mycobacterial envelope is provided with introductions to all of the proteins, lipids, and cofactors involved in this production. The authors need to add a cartoon image of this overall pathway with subsections highlighting the key subpathways that MtInhA is directly involved in. Additionally, the authors should add a chemical scheme showing the chemical reaction performed by MtInhA with chemical structures of the multiple substrates/products. Adding information about the MtInhA mechanism and what is already known about its kinetics in graphical form would also help interpret their data in relation to these past results. A clearer statement of their direct hypothesis would also be helpful. For formatting, the authors should add subheadings to break up these multiple introduction topics.

b. The authors significantly trim back the extraneous background on M. tuberculosis biology, drug treatment, and cell wall synthesis. A one paragraph summary of MtInhA and its role in M. tuberculosis biology would probably suffice. In this shortened version, the authors should still provide the chemical scheme for the MtInhA reaction and a graphical representation of the MtInhA mechanism and summary of past kinetic findings about MtInhA. This will help with interpretation of their results in either format.

Response: As suggested, we, the authors, decided that trimming back the Introduction section would offer a more focused background of our contribution to the readers of Future Pharmacology journal. Note that references were deleted accordingly.

The following sentences were deleted (lines are for the previous version):

Lines 35-40: “The WHO report has also highlighted that TB will rank as the second leading cause of death from a single infectious agent in 2020 and 2021, as COVID-19 ranking first [1]. The number of deaths caused by TB in 2021 (1.4 million) was larger than ones due to HIV/AIDS (0.65 million), and TB mortality was more affected by the COVID-19 pandemic than HIV/AIDS [1]. Although TB disease may affect anyone anywhere, most cases occur in adults.”

Lines 47-49: “This treatment has the additional benefit of curtailing onward transmission of infection. The Global TB Drug Facility supplies a complete 6-month course for about US$ 40 per person.”

Lines 56-68: “The current guidelines of WHO include a 6-month all-oral treatment regimen comprising of bedaquiline, pretomanid, linezolid and moxifloxacin (BPaLM) for patients with MDR/RR-TB where fluoroquinolone susceptibility is presumed or documented (BPaL if fluoroquinolone resistance is confirmed), and a 9-month all-oral regiment comprising of bedaquiline (used for 6 months), in combination wih levofloxacin/moxifloxacin, ethionamide, ethambutol, isoniazid (high dose), pyrazinamide and clofizimine (for 4 months) followed by treatment with levofloxacin/moxifloxacin, clofazimine, ethambutol and pyrazinamide (for 5 months), in which ethionamide can be replaced by 2 months of linezolid [3]. It has been estimated that savings of governments could reach U$740 million annually if all patients infected with either MDR/RR-TB or XDR-RB were to transition to BPaLM/BPal treatment, which could fund MDR/RR-TB treatment for additional 400,000 patients or drug-susceptible TB treatmet for 3.1 million patients [4].”

The following sentence (Page 2, lines 91-93 of current version):

“The elongation of carbon chain of fatty acids occurs through cycles of chemical reactions catalyzed by b-ketoacyl synthase (KAS, condensing enzyme), b-ketoacyl reductase (KAR), b-hydroxyacyl dehydrase (DE), and enoyl reductase (ENR) enzymes [10].”

replaces (lines 109-113 of previous version):

“The elongation of carbon chain of fatty acids occurs through cycles of condensation, b-keto reduction, dehydration and enoyl reduction [12]. These chemical reactions are catalyzed by, respectively, b-ketoacyl synthase (KAS, condensing enzyme), b-ketoacyl reductase (KAR), b-hydroxyacyl dehydrase (DE), and enoyl reductase (ENR) enzymes [12].”

The following sentences were deleted (lines 125-136 of previous version):

“Enzymes, including methyltransferases, modify the meromycolate branch introducing double bonds, cyclopropyl, hydroxy, methoxy, and keto functional groups [13]. This chemically modified meromycolyl-ACP and the a-alkyl-CoA chains are activated by, respectively, FadD32 (a fatty acyl-AMP-ligase) and AccD4 (a carboxyltransferase), yielding meromycolyl-AMP and 2-carboxy acyl-CoA [14]. These activated acyl products are coupled together via Claisen-type condensation by Pks13 (a type I polyketide synthase) producing a-alkyl b-keto thioester, which is released in the form a-alkyl b-ketoacyl by the thioesterase activity of Pks13. Reduction of a-alkyl b-ketoacyl by CmrA yields a-alkyl b-hydroxy fatty acids (mycolic acids) that are translocated by MmpL3 to the periplasm in the form of trehalose mono-mycolate, which are then used as substrates by the mycolyltransferase enymes of the Ag85 complex to be transferred to the arabinogalactan complex or to another trehalose mono-mycolate to form trehalose di-mycolate [13, 14].”

The following sentences (and reference #19) have been added to this revised version of our manuscript (Page 3, lines 127-130):

“A thorough description of chemical and kinetic mechanisms, structural data and inhibitors of MtInhA enzyme activity has recently been reported [19]. The graphical abstract of our contribution strives to both give a summary of past kinetic data and the conclusions reached by the studies here described.”

Note that the reference below was also added to this revised version:

Reference #19: “Hopf, F.S.M.; Roth, C.D.; de Souza, E.V.; Galina, L.; Czeczot, A.M.; Machado, P.; Basso, L.A.; Bizarro, C.V. Bacterial enoyl-reductases: The ever-growing list of Fabs, their mechanisms and inhibition. Front. Microbiol. 2022, 13, 891610. doi: 10.3389/fmicb.2022.891610.”

As suggested, Figure 1 shows the MtInhA-catalyzed chemical reaction (page 3). The remaining figures were hence renumbered. The legend of Figure 1 is as follows (page 3, lines 132-134): “Figure 1. MtInhA catalyzes C-4S hydride transfer from ß-NADH to 2-trans-enoyl ACP(CoA) and protonation of enolate intermediate results in formation of NAD+ and enoyl-ACP(CoA) products. Figure drawn using BioRender.com.“

Note that the following sentence (page3, lines 117-120):

MtInhA (EC 1.3.1.9) enzyme catalyzes the hydride transfer from 4S hydrogen of ß-NADH to carbon-3 of long-chain 2-trans-enoyl thioester substrates (enoyl-ACP or enoyl-CoA), enolate formation followed by protonation yielding NAD+ and acyl (enoyl-ACP or acyl-CoA) products [12, 16, 17] (Figure 1).”

Replaces (page 3, lines 148-150 of previous version):

MtInhA (EC 1.3.1.9) enzyme catalyzes the hydride transfer from 4S hydrogen of ß-NADH to carbon-3 of long-chain 2-trans-enoyl thioester substrates (enoyl-ACP or enoyl-CoA) producing NAD+ and acyl (enoyl-ACP or acyl-CoA) products [16, 20, 21].”

Issue: 2. Similar to the introduction, the conclusion section deviates far from the main story of the results from this article discussing allostery, historical interpretation of allostery, cryptic allosteric sites, computer programs to find cryptic sites, and an unrelated allosteric inhibitor of tryptophan synthase from M. tuberculosis. Unlike the introduction where the extraneous information is mostly harmless to the science presented, this winding conclusion section makes it very difficult to figure out what the major conclusions are from their results. For example, the long discussion of cryptic sites and allosteric inhibitors is irrelevant to MtInhA as the authors state in lines 448-449 that no cryptic sites have been identified in MtInhA. This conclusion section needs to be completely revised and refocused on putting their results in context with the previous kinetic findings. A figure to summarize their results, especially in relation to their new introductory figure summarizing past kinetic findings about MtInhA, would really help understand their kinetic results.

Response: As suggested, the conclusion section has been shortened. The following sentences and publication related to them (reference #34 of previous version) were deleted:

Page 10, lines 417-423 of previous version: “Notwithstanding, it has been found that cryptic sites (sites that form a pocket in a holo structure, but not in the apo structure) tend to be conserved in evolution as traditional binding pockets but are less hydrophobic and more flexible [34]. These authors have constructed a set of structurally defined apo-holo pairs with cryptic sites of the human proteome and these cryptic sites were characterized in terms of their sequence, structure, and dynamics attributes [34]. The resulting CryptoSite Web server was made available by the authors at http://salilab.org/cryptosite.”

Reference #34 (page 13, lines 549-551 of previous version): “Cimermancic, P.; Weinkam, P.; Rettenmaier, T.J.; Bichmann, L.; Keedy, D.A.; Woldeyes, R.A.; Schneidman-Duhovny, D.; Demerdash, O.N.; Mitchell, J.C.; Wells, J.A.; et al. CryptoSite: expanding the druggable proteome by characterization and prediction of cryptic binding sites. J. Mol. Biol. 2016, 428, 709-719. doi: 10.1016/j.jmb.2016.01.029.”

The following sentences were also removed from the current version of our manuscript (page 11, lines 449-454 of previous version): “To the best of our knowledge, there has been no report on allosteric sites for MtInhA. The search for cryptic sites (e.g., not-yet-known allosteric sites) of holo protein structure appears to be worth pursuing to unveil new strategies to be added to the current substrate active site-based efforts to the rational design of MtInhA enzyme inhibitors. In a broader view, these efforts may also contribute to help answer an intriguing proposal that all (nonfibrous) proteins are potentialy allosteric [38].”

Reference #38 (previous version) was deleted accordingly: “Gunasekaran, K.; Ma, B.; Nussinov, R. Is allostery an intrinsic property of all dynamic proteins? Proteins. 2004, 57, 433-443. doi: 10.1002/prot.20232.”

Please, allow me to disagree that discussing allostery and historical interpretation of allostery is not relevant to our manuscript. The data described in our manuscript suggest a complex interplay between different states of free enzyme and enzyme-ligand complexes. Moreover, the data here presented show that allostery indeed plays a role in the mode of action of MtInhA. Whether or not any allosteric site of this enzyme may be unveiled remains to be shown. At any rate, attempts to identify allosteric sites in MtInhA enzyme appears to be worth pursuing as new strategies to inhibit this enzyme activity would thus exploit a larger chemical space than is currently pursued.

We, the authors, also are of opinion that describing an allosteric inhibitor of tryptophan synthase from M. tuberculosis is related to our efforts as it was shown to kill M. tuberculosis. Hence efforts to try to find allosteric sites in MtInhA are worth pursuing as discussed and suggested by the results of single-turnover presented in our manuscript showing that cooperativity play a role in the mode of action of this enzyme.

In our opinion, the results described in our manuscript have been put in context as can be evaluated by the following sentences of this revised version (page 10, lines 417-424) and that were present in the previous version: “This study also highlights the effectiveness of allosteric inhibition for dynamic protein targets that are essential in vivo despite being dispensable under in vitro conditions [30]. The single-turnover results here described suggest that MtInhA displays a typical symmetry model [21] with free enzyme in equilibrium between two conformers, followed by NADH binding to one conformer and conformational change of this binary complex. On the other hand, a sequential model [22] with only one conformer of free MtInhA in solution to which DD-CoA binds with homotropic positive cooperativity was invoked.”

Issue: 3. The presentation of figures in the article and accompanying figure legends need to be revised to make it easier to interpret their experimental results.

a. Figure 1: Both figures are too small to read. Text and plotted points need to be made larger. The figure legend needs significant expansion. What data are presented in this figure? What substrate is tested? What are the experimental values for enzyme and substrate concentrations used in this experiment? What is the line shown on the major figure? A more complete descriptive figure legend needs to be added.

Response: Indeed. Figure 2 (previous Figure 1) was edited as suggested. Figure 2 was expanded as requested. We have found appropriate to divide this figure into two panels (right and left) to make data easily seen. The legend is now as follows: “Figure 2. Left panel: representative stopped-flow trace of MtInhA (15 µM) reaction showing decrease in absorbance at 340 nm at 25 °C upon conversion of NADH (14 µM) and DD-CoA (225 µM) into NAD+ and dodecanoyl-CoA over a course period (left panel). The pre-steady-state time course was fitted to a single exponential decay equation (Equation 2). Note: All concentrations values for enzyme and substrates are for mixing chamber concentrations. Right panel: represents the calculated residual values plotted against time (s) using Equation 2.”

b. Figure 2: Text should be made larger. For the figure legend, what type of error is shown? What are the experimental conditions?

Response: Thanks for bringing it to our attention. Labels (texts) of Y- and X-axis of Figure 3 (previous Figure 2) were enlarged. The data given in Figure 3 were derived from the best fit of the data of Figure 2 to a single-exponential function, which also yielded the standard error (or sample standard deviation from the mean value) shown in Figure 3. The Figure 3 legend is now as follows: “Figure 3. Linear dependence of kobs values on the degree of occupancy of active sites of MtInhA (15 µM) without pre-incubation with NADH concentrations of 5 µM (28% occupancy), 10 µM (52% occupancy) and 14 µM (66% occupancy). The kobs values were derived from the best fit of the data of Figure 2 to a single-exponential function, yielding the standard error represented by the bars in the plot.” The experimental conditions were as described for Figure 2, and we deemed appropriate not to include them in Figure 3 legend as it would have been redundant.

c. Figure 3: Text and symbols should be made larger. Different colors for the two different data sets would more clearly differentiate them. What type of error is plotted? What are the experimental conditions?

Response: As suggested for Figure 4 (previous Figure 3), text and symbols were made larger. The two data set are now coloured. The errors shown in Figure 4 are as described in response to the previous issue raised by this reviewer and repeated here. A short description of the experimental conditions is now provided. Accordingly, Figure 4 legend is now as follows: “Figure 4. The dependence of kobs values derived from data fitting to a double-exponential function (Equation 3) on the degree of occupancy of active sites of MtInhA (15 mM) pre-incubated with NADH concentrations of 5 mM (28% occupancy), 10 mM (52% occupancy) and 14 mM (66% occupancy). The (▲) represents the kobs1 for the fast-phase of the biphasic reaction (in black), and (●) represent kobs2 for the slow-phase of the biphasic reaction (in blue). The bars in the plot represent the standard errors of data fitting to Equation 3.”

d. Figure 5: Add color to more clearly differentiate the various conditions. Add experimental details about the conditions used in this experiment.

Response: As suggested for Figure 6 (previous Figure 5), the stopped-flow traces are shown in different colours. A summary of the experimental conditions are given in the figure legend, which is now as follows: “Figure 6. Stopped-flow trace for substrate consumption by measuring the decrease in absorbance at 340 nm at 25 °C for MtInhA (10 µM). Data were fitted to Equation 2 (single exponential decay). The upper time course (in black) is for negative chemical control in which 100 mM Pipes pH 7.0 buffer in one syringe was mixed with NADH (600 μM) and DD-CoA (450 μM) in another syringe (absence of enzyme). The lower time course (in green) is for mixing enzyme (20 µM) in one syringe mixed with 100 mM Pipes pH 7.0 buffer in another syringe (absence of substrates). The time course shown in blue is for the chemical reaction upon mixing MtInhA (20 µM) in one syringe with NADH (600 μM) and DD-CoA (450 μM) in another syringe. Note that the mixing chamber concentrations are 10 µM MtInhA, 300 µM NADH, and 225 µM DD-CoA”

e. Building on the introductory figure summarizing past kinetic analyses of MtInhA, the authors should add a cartoon figure into their results section summarizing their various hypotheses about MtInhA structure and conformational change and how to interpret these hypotheses in relation to their experimental data.

Response: We, the authors, are of opinion that the Graphical Abstract gives a pictorial description summarizing our conclusions. Moreover, it also gives a summary of past data on MtInhA enzyme (steady-state and pre-steady-state) kinetics.

Issue: 4. Add some more background descriptions to the results and discussion section

Response: We, the authors, are of opinion that efforts were made to give sufficient background to discuss the results described in our manuscript.

a. How did the authors decide on a quadratic equation (eq 1) for the binding sites on MtInhA? The authors do not justify this choice.

Response: Please, forgive me if I have misunderstood this issue. The quadratic equation (Equation 1) was employed to give, for instance, the MtInhA:NADH binary complex ([ES]) concentration using the overall dissociation constant value of 2 mM (as described in the Results and Discussion section), and mixing 15 µM of tetramers of MtInhA with either 5, 10 or 14 µM of NADH (as described in the Materials and Method section). Hence, as the quadratic equation gives the concentration of the binary complex, it is just a matter of dividing the binary complex concentration by total enzyme concentration and multiplying by 100 {([MtInhA:NADH]/[MtInhA])x100}. This is standard practice between scientists working on enzyme kinetics.

b. How did the authors decide on their concentrations of NADH to bind to varying occupancies of monomers? (Lines 246-247). The origination of these values is unclear.

Response: This was based on the dissociation constant value of 2 mM (KD = 2 µM) reported by Quémard, A.; Sacchettini, J.C.; Dessen, A.; Vilchèze, C.; Bittman, R.; Jacobs, W.R.Jr.; Blanchard, J.S. Enzymatic characteriza-tion of the target for isoniazid in Mycobacterium tuberculosis. Biochemistry. 1995, 34, 8235-8241. doi: 10.1021/bi00026a004 (reference #12) and the MtInhA enzyme concentration of 15 µM (tetramer). The quadratic equation can thus be employed fixing the concentration of the binary complex [ES] to be, for instance, ¼ of the enzyme concentration and calculate the resulting the concentration of NADH that satisfy these values.

c. Similarly in Table 1, the authors present calculations for the percent of active sites occupied. Where do these values come from? Are these calculated values or are they experimentally determined?

Response: This issue has been dealt with in response to the two ones raised above.

d. Lines 284-285: How do the authors come to the conclusion about the isomerization step and DD-CoA binding based on the data presented in Figure 3? The authors need to explain this data interpretation more thoroughly.

Response: The experimental results for pre-incubation of enzyme with NADH and mixing with DD-CoA yielded a double-exponential trace. Fitting the data to a double exponential function yelded values for kobs1 and kobs2. The decreasing kobs1 values as a function of increasing NADH concentration (now Figure 4) are interpreted as an isomerization step between two forms of MtInhA:NADH binary complex in solution and binding of DD-CoA to one of the conformers. This conclusion is standard for enzymologists that work with pre-steady state kinetics. The following textbook may be consulted to verify the mathematical formulation and explanations for this type of substrate dependence: Hiromi, K. Kinetics of fast enzyme reactions, Halsted Press: New York, U.S.A., 1979; pp. 187-253. The following reference (#20 of the current version) can also be consulted: Oliveira, J.S.; Pereira, J.H.; Canduri, F.; Rodrigues, N.C.; de Souza, O.N.; Azevedo W.F.Jr.; Basso, L.A.; Santos, D.S. Crystallographic and pre-steady-state kinetics studies on binding of NADH to wild-type and isoniazid-resistant enoyl-ACP(CoA) reductase enzymes from Mycobacterium tuberculosis. J. Mol. Biol. 2006, 359, 646-666. doi: 10.1016/j.jmb.2006.03.055. Incidentally, this reference has been indicated in the current version of our manucript in case the readers need further explanations. Now it is hence as follows (lines 307-309): “These data indicate an isomerization step between two forms of MtInhA:NADH binary complex in solution and binding of DD-CoA to one of the conformers [20].”

Issue: 5. Minor details

a. Multiple spelling and grammatical errors in introduction. Please check carefully.

Response: Spelling and grammatical mistakes were corrected in the Introduction section.

b. Reference 1 is blank.

Response: Thanks for bringing it to our attention. It has been corrected.

Reviewer 3 Report

"Single turnover of transient of reactants support a complex 2 interplay of conformational states in the mode of action of 3 Mycobacterium tuberculosis enoyl reductase" is a primary article about the M. tuberculosis enzyme InhA. It describes the authors' experimental setting and results to evaluate some kinetics values of this allosteric enzyme, some usefull information for development of novel inhibitors.

The article is well written and the resoning is intersting. I recommend this article for the publication on Future Pharmacology.

Following a couple of hints to improve the manuscript:

line 197: please check the sentence before the fullstop, wonder if there is an unwanted  "the". 

figure 3, line 295: in Kobs, "obs" is not subscripted. Not sure if it is wanted since you're talking of Kobs in general or just a distraction, since in the text (es: line 311) obs is subscripted.

Author Response

REVIEWER #3

We, the authors, were glad that reviewer #3 recommended “this article for the publication on Future Pharmacology.” There were, however, a few changes suggested by this reviewer to be made to our manuscript. Our responses to these suggestions (issues) are as follows:

Issue #1: “line 197: please check the sentence before the fullstop, wonder if there is an unwanted "the".

Response: Please, forgive me if I misunderstood your suggestion. In my opinion, there is no unwanted “the” before the full stop. The word “the” was employed at the beginning of the two models (symmetry and sequential) as they are known and further details about them are given in the subsequent sentences.  

Issue #2: “figure 3, line 295: in Kobs, "obs" is not subscripted. Not sure if it is wanted since you're talking of Kobs in general or just a distraction, since in the text (es: line 311) obs is subscripted.”

Response: Thanks for bringing it to our attention. The observed rate constants are now subscripted in Figure 4 of current version (Figure 3 of previous version).

Round 2

Reviewer 2 Report

In this revised version of the manuscript, the authors made some inroads to the changes indicated by this reviewer to the original article. After rereading the article with these revisions, this reviewer does appreciate the grammatical corrections to the introduction, the single figure showing the overall reaction, and the significantly expanded figure legends. However, this reviewer is still of the opinion that the article would be significantly improved with the outlined edits below. Many of these are identical to the first revision. Additionally, the authors responded back to other comments within the reviewer feedback but did not work these explanations into the written article. Incorporating these explanations into the article would help clarify some of these issues. The article still reads more like a review article, especially the background, with only minor new results presented, which makes it difficult to follow the essential background necessary to interpret the results and to understand where these results fit into previous studies. The authors still need to either add more figures and formatting structure to help organize the extraneous background material or they need to much more substantially pare down the article to focus on only the relevant background. With more complete implementation of the revisions outlined below, the article could be ready for publication in Future Pharmacology.

1.     The introduction reads more like a review article over M. tuberculosis biology, drug treatment, cell wall synthesis, and eventually MtInhA mechanism and kinetics than a focused background to this article. This reviewer sees two potential outcomes to this current format. The authors need to choose one of these options and fully implement this option into the introduction.

a.     The authors keep this amount of background and add some figures and formatting to help the reader follow along with this review material. Specifically, multiple figures should be added to the background to support the written description. For example, a complete rundown of the mycobacterial envelope is provided with introductions to all of the proteins, lipids, and cofactors involved in this production. The authors need to add a cartoon image of this overall pathway with subsections highlighting the key subpathways that MtInhA is directly involved in. Adding information about the MtInhA mechanism and what is already known about its kinetics in graphical form would also help interpret their data in relation to these past results. A clearer statement of their direct hypothesis would also be helpful. For formatting, the authors should add subheadings to break up these multiple introduction topics.

b.     The authors significantly trim back the extraneous background on M. tuberculosis biology, drug treatment, and cell wall synthesis. A one paragraph summary of MtInhA and its role in M. tuberculosis biology would probably suffice. In this shortened version, the authors should still add a thoughtful graphical representation of the MtInhA mechanism and summary of past kinetic findings about MtInhA. This will help with interpretation of their results in either format.

2.     The conclusion is improved in the revised version and I can understand the authors point of view about the inclusion of information about allostery and Trp synthase.

3.     Good revisions to the figures and figure legends. Comprehension of the results in the article would still be improved with the addition of a summarizing cartoon figure.

a.     Building on the introductory figure summarizing past kinetic analyses of MtInhA, the authors should add a cartoon figure into their results section summarizing their various hypotheses about MtInhA structure and conformational change and how to interpret these hypotheses in relation to their experimental data.

4.     Add some more background descriptions to the results and discussion section. This is the section where the authors should use the explanations provided to this reviewer in the reviewer response and work these explanations directly into the article. The authors assumption that these topics are common knowledge is not justified as this reviewer was not previously aware of many of these details.

a.     How did the authors decide on a quadratic equation (eq 1) for the binding sites on MtInhA? The authors do not justify this choice.

b.     How did the authors decide on their concentrations of NADH to bind to varying occupancies of monomers? (Lines 246-247). The origination of these values is unclear.

c.     Similarly in Table 1, the authors present calculations for the percent of active sites occupied. Where do these values come from? Are these calculated values or are they experimentally determined?

d.     Lines 284-285: How do the authors come to the conclusion about the isomerization step and DD-CoA binding based on the data presented in Figure 3? The authors need to explain this data interpretation more thoroughly.

5.     Good revisions to minor grammatical issues.  

Author Response

Comments and Suggestions for Authors

In this revised version of the manuscript, the authors made some inroads to the changes indicated by this reviewer to the original article. After rereading the article with these revisions, this reviewer does appreciate the grammatical corrections to the introduction, the single figure showing the overall reaction, and the significantly expanded figure legends. However, this reviewer is still of the opinion that the article would be significantly improved with the outlined edits below. Many of these are identical to the first revision. Additionally, the authors responded back to other comments within the reviewer feedback but did not work these explanations into the written article. Incorporating these explanations into the article would help clarify some of these issues. The article still reads more like a review article, especially the background, with only minor new results presented, which makes it difficult to follow the essential background necessary to interpret the results and to understand where these results fit into previous studies. The authors still need to either add more figures and formatting structure to help organize the extraneous background material or they need to much more substantially pare down the article to focus on only the relevant background. With more complete implementation of the revisions outlined below, the article could be ready for publication in Future Pharmacology.

Issue 1. The introduction reads more like a review article over M. tuberculosis biology, drug treatment, cell wall synthesis, and eventually MtInhA mechanism and kinetics than a focused background to this article. This reviewer sees two potential outcomes to this current format. The authors need to choose one of these options and fully implement this option into the introduction.

  1. The authors keep this amount of background and add some figures and formatting to help the reader follow along with this review material. Specifically, multiple figures should be added to the background to support the written description. For example, a complete rundown of the mycobacterial envelope is provided with introductions to all of the proteins, lipids, and cofactors involved in this production. The authors need to add a cartoon image of this overall pathway with subsections highlighting the key subpathways that MtInhA is directly involved in. Adding information about the MtInhA mechanism and what is already known about its kinetics in graphical form would also help interpret their data in relation to these past results. A clearer statement of their direct hypothesis would also be helpful. For formatting, the authors should add subheadings to break up these multiple introduction topics.

  1. The authors significantly trim back the extraneous background on M. tuberculosis biology, drug treatment, and cell wall synthesis. A one paragraph summary of MtInhA and its role in M. tuberculosis biology would probably suffice. In this shortened version, the authors should still add a thoughtful graphical representation of the MtInhA mechanism and summary of past kinetic findings about MtInhA. This will help with interpretation of their results in either format.

Response: We, the authors, decided to choose trimming back further the Introduction section as one of the options suggested by this reviewer. Accordingly, “extraneous background on M. tuberculosis biology, drug treatment, and cell wall synthesis” were excluded. Note that references were deleted accordingly.

The following sentence (line 37 of current version):

“Moreover, drug-resistance continues to be a public health threat [1].”

replaces (lines 37-39 of previous version):

“Drug-resistance continues to be a public health threat as there were an estimated 450,000 cases of patients developed rifampicin-resistant TB (RR-TB) or multidrug-resistant TB (MDR-TB).”

The following sentences were deleted (lines 43-48 of previous version):

“Global average treatment success rate for RR-TB is 60%. Although the current guidelines for treatment of pan-sensitive TB have been established 35 years ago, a number of clinical trials are underway not only to treat pan-sensitive but also MDR-TB, extensively drug-resistant TB (XDR-TB: TB that is resistant to rifampicin, plus any fluoroquinolone, plus at least one of the drugs bedaquiline and linezolid) and latent TB [2].”

The word “even” was removed from the following sentence (page 44 of current version): “...needed to reduce even further the course...” Ii is thus now as follows: “...needed to reduce further the course...”

The following sentence (lines 80-82 of current version):

“The latter are high-molecular-weight a-alkyl, b-hydroxy fatty acids that are covalently attached to hexaarabinosyl of non-reducing termini of arabinanlactan polymer in the mycobacterial cell wall [9, 10].”

now replaces (lines 80-83 of previous versin):

The latter are high-molecular-weight a-alkyl, b-hydroxy fatty acids that are covalently attached to hexaarabinosyl of non-reducing termini of arabinanlactan polymer, forming clusters of tetramycolylpentaarabinosyl in the mycobacterial cell wall [9, 10].

The following sentence was deleted (82-84 of previous version):

“The pyrolytic cleavage of mycolic acids releases alpha fatty acid (a branch) and beta aldehyde (b-meroaldehyde), which is usually referred to as the meromycolate branch [10].”

The following sentence was deleted (lines 82-86):

“The elongation of carbon chain of fatty acids occurs through cycles of chemical reactions catalyzed by b-ketoacyl synthase (KAS, condensing enzyme), b-ketoacyl reductase (KAR), b-hydroxyacyl dehydrase (DE), and enoyl reductase (ENR) enzymes [10]. There are two types of fatty acid synthase systems (FAS) that catalyze these chemical reactions [8, 9, 10].”

Rephrasing was hence needed (lines 82-84):

“There are two types of fatty acid synthase systems (FAS) that catalyze b-ketoacyl synthesis, b-ketoacyl reduction, b-hydroxyacyl dehydration and enoyl reduction chemical reactions [8, 9, 10].”

Please, allow me to disagree with sentences concerning the remaning cell wall synthesis in mycobacteria. We have kept sentences discussing cell wall synthesis that we deemed needed to put MtInhA enzyme activity in context. We are of opinion that they are not “extraneous background”.

As already dealt with in response to issue #1 raised by this reviewer in the first round, the following sentences (and reference #19) were already added to our manuscript (lines 116-119 of current version):

“A thorough description of chemical and kinetic mechanisms, structural data and inhibitors of MtInhA enzyme activity has recently been reported [19]. The graphical abstract of our contribution strives to both give a summary of past kinetic data and the conclusions reached by the studies here described.”

We, the authors, are of opinion that both Figure 1 and the Graphical Abstract gives a “thoughtful graphical representation of the MtInhA mechanism and summary of past kinetic findings about MtInhA”. Moreover, the Graphical Abstract gives a summary of past data on MtInhA enzyme (steady-state and pre-steady-state) kinetics and the conclusions arrived by interpreting the single-turnover data described in our manuscript.

Issue 2. The conclusion is improved in the revised version and I can understand the authors point of view about the inclusion of information about allostery and Trp synthase.

Response: We, the authors, were pleased that reviewer #2 agreed that inclusion of an example of an allosteric inhibitor of tryptophan synthase was appropriate.

Issue 3. Good revisions to the figures and figure legends. Comprehension of the results in the article would still be improved with the addition of a summarizing cartoon figure.

  1. Building on the introductory figure summarizing past kinetic analyses of MtInhA, the authors should add a cartoon figure into their results section summarizing their various hypotheses about MtInhA structure and conformational change and how to interpret these hypotheses in relation to their experimental data.

Response: We are of opinion that this issued has been dealt with in response to issue #1 raised by the reviewer in this second round of comments/suggestions. Moreover, trying to assign changes in enzyme conformations detected by the single turnover experiments to any particular MtInhA tridimensional structure is not warranted. 

Issue 4. Add some more background descriptions to the results and discussion section. This is the section where the authors should use the explanations provided to this reviewer in the reviewer response and work these explanations directly into the article. The authors assumption that these topics are common knowledge is not justified as this reviewer was not previously aware of many of these details.

  1. How did the authors decide on a quadratic equation (eq 1) for the binding sites on MtInhA? The authors do not justify this choice.

Response: The following sentences were added to the current version of our manuscript:

Lines 250-252: “Equation 1 represents the solution of a quadratic equation for a simple binding process [25].”

Lines 253-263: “Equation 1 was employed to give, for instance, the MtInhA:NADH binary complex ([ES]) concentration using the overall dissociation constant value of 2 mM (as described in the Results and Discussion section), and mixing 15 µM of tetramers of MtInhA with either 5, 10 or 14 µM of NADH (as described in the Materials and Method section). Hence, as the quadratic equation gives the concentration of the binary complex, it is just a matter of dividing the binary complex concentration by total enzyme concentration and multiplying by 100 {([MtInhA:NADH]/[MtInhA])x100}. The NADH concentrations were chosen to obtain increasing number of monomers of tetrameric MtInhA enzyme active sites occupied by this substrate (KD = 2 µM) [12]. Namely, 5 µM equals to one monomer, 10 µM to two monomers and 14 µM to three monomers of tetrameric MtInhA occupied by NADH (Table 1).”

Note that reference 25 was added:

“[25] Birdsall, B.; King, R.W.; Wheeler, M.R.; Lewis, C.A.Jr.; Goode S.R.; Dunlap, R.B.; Roberts, G.C. Correction for light absorption in fluorescence studies of protein-ligand interactions. Anal. Biochem. 1983, 132, 353-361. doi: 10.1016/0003-2697(83)90020-9.”

The remaining references were renumbered accordingly.

  1. How did the authors decide on their concentrations of NADH to bind to varying occupancies of monomers? (Lines 246-247). The origination of these values is unclear.

Response: This issue has been dealt with in response to the previous issue.

  1. Similarly in Table 1, the authors present calculations for the percent of active sites occupied. Where do these values come from? Are these calculated values or are they experimentally determined?

Response: This issue has already been dealt with (see above).

  1. Lines 284-285: How do the authors come to the conclusion about the isomerization step and DD-CoA binding based on the data presented in Figure 3? The authors need to explain this data interpretation more thoroughly.

Response: Please, forgive me if I misunderstood this issue. The sentences related to Figure 3 of the current version of our manuscript are as follows (lines 282-290 of previous version):

Lines 280-288 of current version: “As pointed out above, single turnover experiments can be used to test whether the sites of oligomeric enzymes are kinetically identical. The results for the single-turnover experiment without incubation show a linear relationship between kobs values and increasing percentage of MtInhA sites occupied by NADH (Figure 3). These results suggest that the active sites of tetrameric MtInhA are kinetically equivalent, which is also borned out by the single exponential records for any degree of site occupancy. However, it is also possible that NADH binding occurs after DD-CoA substrate binding to form the ternary complex and ensuing turnover as there was no pre-incubation of enzyme with reduced dinucleotide substrate.”

Accordingly, no isomerization step was invoked based on the results presented in Figure 3.

Issue 5. Good revisions to minor grammatical issues.  

Reponse: Spelling, grammar and sentence strucuture were edited as needed throughout.

Incidentally, the following reference has been deleted: “Pauli, I.; dos Santos, R.N.; Rostirolla, D.C.; Martinelli, L.K.; Ducati, R.G.; Timmers, L.F.; Basso, L.A.; Santos, D.S.; Guido, R.V.; Andricopulo, A.D.; Norberto de Souza, O. Discovery of new inhibitors of Mycobacterium tuberculosis InhA enzyme using virtual screening and a 3D-pharmacophore-based approach. J. Chem. Inf. Model. 2013, 53, 2390-2401. doi: 10.1021/ci400202t.”
